# Microstructural Evolution and Stability of Coarse-Grained S31254 Super Austenitic Stainless Steel during Hot Deformation

Jia Xing [1,2,*], Chengzhi Liu [2], Aimin Li [3], Shouming Wang [3], Xinjie Zhang [3] and Yongxin Shi [3]

1   School of Mechanical Engineering, North University of China, Taiyuan 030051, China
2   Institute of Special Metal Materials and Equipment, North University of China, Taiyuan 030051, China
3   Avic Shangda Superalloys Company Limited, Xingtai 054800, China
*   Correspondence: xingjia@nuc.edu.cn

**Abstract:** The ingot of S31254 super austenitic stainless steel (SASS) was annealed at 1220 °C for 70 h to eliminate the segregation of Mo element, and the grain size grows to the level of millimeter. The stress–strain response and microstructural evolution of coarse-grained S31254 SASS were investigated by hot compression tests in the temperature range of 950–1250 °C and strain rate range of 0.001–10 s$^{-1}$. The results showed that the energy required for plastic deformation improved with the increase of strain rate and the decrease of deformation temperature. The hot deformation activation energy was calculated to be 542.91 kJ·mol$^{-1}$ through the regression analysis of hyperbolic-sine function, and the constitutive equation was established. Processing maps were constructed, and two optimal hot working parameters ranges were clarified. Due to the low fraction of grain boundaries, the main deformation mechanism of coarse-grained S31254 SASS was dynamic recovery. However, when the deformation temperature improved to 1250 °C, recrystallized grains began to nucleate and grow along with the band-like structure within the austenitic grains. When the deformation temperature is 950–1150 °C, the microstructural stability of S31254 SASS under tension stress was excellent. However, when the temperature and the strain rate were 1250 °C and 0.5 s$^{-1}$ respectively, the microstructural stability deteriorated resulting from the formation of δ-ferrite phase and local melting of austenitic grain boundaries.

**Keywords:** super austenitic stainless steel; coarse-grained austenite; constitutive model; processing map; hot tensile; dynamic recrystallization

## 1. Introduction

Super austenitic stainless steel (SASS) has more excellent corrosion resistance and mechanical properties than common austenitic stainless steel. Because its cost of raw material is lower than that of nickel-based alloys, it can be regarded as a cheap substitute in many harsh environments [1]. According to the development tendency, SASS will have greater application prospects in the future.

The properties of steel are controlled by microstructure characteristics [2,3]. It has been widely recognized that the SASS can achieve grain refinement by controlling the process of hot working to obtain excellent performance [4–6]. Up to now, numerous studies have paid attention to the microstructure evolution of SASS during hot deformation. Han et al. investigated the deformation behavior and microstructure evolution of S31254 SASS in hot compression tests with the deformation temperature ranging from 900–1200 °C and strain rate ranging from 0.01–10 s$^{-1}$. The results presented the dynamic recrystallization nucleated mainly through grain boundary bulging phenomenon [7,8]. Pu et al. established the hot deformation constitutive equation and processing map of S32654 SASS according to the results of hot compression tests. They attributed the excellent hot plastic properties to the small and uniform size of the recrystallized grains [9]. Momeni et al. studied the carbide

precipitation behavior and its effect on dynamic recrystallization of 1.4563 SASS during hot compression. They confirmed that the $(Cr, Fe, Mo)_{23}C_6$ carbides, which precipitated along with grain boundaries at low temperature and moderate strain rates, inhibited the progress of recrystallization [10].

So, some strategies have been proposed to prepare the SASS materials with fine-grained structures by adjusting hot deformation conditions. However, these strategies face severe challenges in practical applications. That is, the SASS contains a high content of the Mo element. Mo has a strong segregation tendency, which is easy to distribute unevenly in dendritic as-cast microstructure [11]. To eliminate the element segregation, it is necessary to carry out long-term homogenizing of the ingot. However, the long-term homogenizing will inevitably lead to a sharp increase in grain size. In most studies about the dynamic microstructural evolution of SASS, including the above references, the heat treatment duration before deformation was short (less than one hour). So, their initial grain sizes of specimens were fine. For coarse-grained SASS, the microstructural characteristics during hot deformation must differ from those of fine-grained steel. However, few works reported the hot deformation behavior of the coarse-grained SASS. Hermant et al. discussed the microstructural evolution of 316Nb stainless steel with the grain size greater than 130 μm in high-temperature torsion experiments. They suggested that the initial grain size had little effect on the resistance of viscoplastic flow. The increase of the heat treatment temperature could suppress dynamic recrystallization by restricting the bulging of grain boundary [12]. Rehrl et al. investigated the influence of initial grain size on the dynamic recrystallization behavior of 316LN stainless steel. The results showed that the increase in grain size would reduce the storage energy. The insufficient driving force of dynamic recrystallization led to difficulty in grain refinement [13]. Sha et al. studied the effect of coarse-grained austenite on dynamic and static recrystallization behavior in Nb-V-Ti low alloy steel. They found that the ratio between critical strain and peak strain in the coarse-grained specimen was consistent with that in the fine-grained one. The strain-induced precipitation behavior would be inhibited once the deformation temperature exceeded 1000 °C [14]. In summary, the study on the hot deformation behavior of coarse-grained SASS was few. Research about the characteristics of microstructural transformation and the evolution of mechanical properties of coarse-grained S31254 SASS during hot deformation is even more lacking. The evaluation of the structural stability of coarse-grained S31254 SASS forged workpieces was still scarce.

In this article, a commercial S31254 SASS was chosen as the research object. The characteristics of the stress–strain response of the specimens were obtained through hot compression tests, and the microstructural evolution during hot deformation was analyzed. The hot deformation activation energy was calculated and the Arrhenius type constitutive equation was established. The strength and plastic properties at different strain rates were studied by hot tensile tests. The structural stability of the specimen was evaluated by combining the processing maps with the reduction of area in tensile test.

## 2. Experimental Procedures

The chemical composition of S31254 SASS in this experiment is shown in Table 1. The experimental steel was prepared by vacuum induction-melting and poured into several $\Phi$ 430 × 3000 mm ingots. The ingots were annealed at 1220 °C for 70 h to eliminate element segregation and dissolve the brittle σ-phase. After annealing, the size of the austenitic grain grows to several millimeter. The cylindrical specimens were machined from the annealed ingot with a height of 12 mm and diameter of 8 mm. The compression tests were conducted on a Gleeble 3800 thermal and mechanical simulator (Dynamic Systems Inc., New York, NY, USA) in the temperature range of 950–1250 °C and the strain rate range of 0.001–10 s$^{-1}$. To reduce the friction effect during the compression, the upper and lower surfaces of the specimen need to be polished. In addition, the graphite lubricant and tantalum sheet were placed between the specimen and anvil. The specimens were first heated to 1250 °C at a rate of 10 °C/s and held at this temperature for 300 s to relieve

the heat stress. Then, the specimens were cooled to the deformation temperature at a rate of 5 °C/s and held for 120 s to relieve the temperature gradient. The maximum true strain in the tests was 0.7. After the compression, the specimens were quenched in water to retain their microstructural characteristics. During the deformation process, the system automatically recorded experimental data including true stress, true strain, and deformation temperature.

**Table 1.** Chemical composition of the S31254 SASS (in wt%).

| C | Cr | Ni | Mo | Mn | Al | Cu | Si | P | S |
|---|---|---|---|---|---|---|---|---|---|
| 0.011 | 19.74 | 18.12 | 6.13 | 0.46 | 0.032 | 0.64 | 0.30 | 0.024 | 0.001 |

The compressed specimens were sectioned parallel to the compression axis and prepared for microstructure investigation. The observed surfaces were ground and polished, then etched in the solution of aqua regia. The microstructure was characterized by Leica DM1750M metallographic microscope (Leica Microsystems CMS GmbH, Wetzlar, Germany) and Zeiss Sigma-300 field emission scanning electron microscope (Carl Zeiss AG, Oberkochen, Germany). The element distribution was detected by EDX analysis. The deformed microstructure in different conditions was analyzed by EBSD with the working distance of 8.5 mm and the acceleration voltages of 20 kV. The scanning step was 0.8 μm.

The hot tensile tests were also conducted on the Gleeble3800 thermal and mechanical simulator, and the specimen size was $\Phi$ 10 × 121.5 mm. The specimens were first heated to 1250 °C at a rate of 10 °C/s, kept for 300 s, then cooled down to the testing temperature (950, 1050, 1150, 1250 °C) at a rate of 5 °C/s. The experiments started after holding for 30 s at the testing temperatures. The strain rates were 0.5 s$^{-1}$ and 5 s$^{-1}$. Its purpose was to simulate the actual strain rate of hot working.

The melting temperature of the experimental steel was detected by differential scanning calorimetry (DSC) at a heating rate of 20 °C/s from room temperature to 1450 °C. A NETZSCH STA 449 F3 thermal analyzer (Erich NETZSCH GmbH & Co. Holding KG, Selb, Germany) was used in a purified argon gas atmosphere at flowing rate of 50 mL/min. Thermodynamic calculations of the phase composition were carried out using the JMat Pro 7.0 software (Sente Software, Guildford, UK).

## 3. Model and Equation

### 3.1. The Establishment of Arrhenius Type Constitutive Equation

Activation energy can be used as a quantitative index for evaluating the ease of deformation of metals at high temperatures. It can be determined by using the hyperbolic-sine function of the Arronius type constitutive model, which effectively explains the correlation between flow stress, strain rate, and deformation temperature [15,16]. Its expression is:

$$\dot{\varepsilon} = A[\sinh(\alpha\sigma)]^n exp\left(\frac{-Q}{RT}\right) \tag{1}$$

where $\dot{\varepsilon}$ is the strain rate (s$^{-1}$), $A$ is the material constant (s$^{-1}$), $\alpha$ is the stress multiplier (MPa$^{-1}$), $\sigma$ is the flow stress (MPa), $n$ is the stress exponent, $Q$ is the activation energy (J·mol$^{-1}$), $R$ is the gas constant (8.314 J·mol$^{-1}$·K$^{-1}$), and $T$ is the absolute deformation temperature (K) [7]. The activation energy is correlated with the hot deformation mechanism of the metal, such as dynamic recovery, dynamic recrystallization, and the movement of grain boundary, through the $Z$ parameter, and its expression is as follows [17]:

$$Z = \dot{\varepsilon} exp\left(\frac{Q}{RT}\right) \tag{2}$$

After taking the logarithm of both sides of Equations (1) and (3) can be derived as:

$$\log \dot{\varepsilon} + \frac{Q}{2.3R}\frac{1}{T} = n \log[\sinh(\alpha\sigma)] + \log A \tag{3}$$

For a constant deformation temperature, the stress exponent $n$ can be expressed in terms of Equation (4) by partial differentiating Equation (3):

$$n = \left.\frac{\partial \log \dot{\varepsilon}}{\partial \log[\sinh(\alpha\sigma)]}\right|_T \tag{4}$$

Similarly, if the strain rate $\dot{\varepsilon}$ is a specific value, the expression of the activation energy $Q$ can be obtained by taking the partial differentiating Equation (3):

$$Q = 2.3nR\left.\frac{\partial \log[\sinh(\alpha\sigma)]}{\partial\left(\frac{1}{T}\right)}\right|_{\dot{\varepsilon}} \tag{5}$$

To simplify the calculation process, the flow stress usually adopts the value of the peak stress. Not only because the peak stress is a commonly used mechanical property index in the manufacturing industry, but also because the peak stress can be quickly and accurately obtained from the flow curve [18]. If the values of $n$ and $Q$ are required, the value of the stress multiplier $\alpha$ needs to be determined first. Here, a method verified by Uvira and Jonas is adopted [19]. This method makes the value of $\alpha$ vary within a small range of 0.005–0.02 MPa$^{-1}$, and then the value of $n$ will change accordingly. When the variational value of $n$ at each temperature has the most minor standard deviation, the corresponding value of $\alpha$ can be considered the most suitable stress multiplier. The value of $\alpha$ was assigned to 0.012 MPa$^{-1}$ here according to a reported work, because it has been proved to have good representativeness for S31254 SASS [20].

### 3.2. The Establishment of the Processing Maps

According to the dynamic materials model, a deformed workpiece at high temperature can be treated as a dissipater of power [21]. The power dissipation depends on the flow behavior during the processing. The flow stress, which obeys the power law equation, is given by:

$$\sigma = k\cdot\dot{\varepsilon}^m \tag{6}$$

where $\sigma$ is the flow stress (MPa), $\dot{\varepsilon}$ is the strain rate (s$^{-1}$), $m$ is the strain rate sensitivity, and $k$ is the material coefficient. The instantaneous power $P$ could be separated into two complementary parts, $G$ and $J$, with the following function [22]:

$$P = \sigma\cdot\dot{\varepsilon} = G + J = \int_0^{\dot{\varepsilon}} \sigma\cdot d\dot{\varepsilon} + \int_0^{\sigma} \dot{\varepsilon}\cdot d\sigma \tag{7}$$

where $G = \int_0^{\dot{\varepsilon}} \sigma\cdot d\dot{\varepsilon}$, $J = \int_0^{\sigma} \dot{\varepsilon}\cdot d\sigma$.

In Equation (7), $G$ represents the power content, which is the power dissipated during plastic deformation. Most of $G$ is converted into heat, and the remaining is stored in the form of lattice defects [23]. $J$ represents the power co-content, which is related to the microstructural evolution in the process of material deformation, such as dynamic recovery and dynamic recrystallization. The partitioning of power between $J$ and $G$ is affected by the strain rate sensitivity $m$ as follows:

$$m = \frac{\partial J}{\partial G} = \frac{\dot{\varepsilon}\cdot\partial\sigma}{\sigma\cdot\partial\dot{\varepsilon}} = \frac{\partial(\ln\sigma)}{\partial(\ln\dot{\varepsilon})} \tag{8}$$

Based on Equations (7) and (8), when the deformation temperature and strain are constant, $J$ can be expressed as follows:

$$J = \int_0^{\sigma} \dot{\varepsilon}\cdot d\sigma = \frac{m}{m+1}\sigma\dot{\varepsilon} \tag{9}$$

If $m = 1$, the material is an ideal linear dissipater, and $J$ gets the maximum value, $J_{max}$. Thus,

$$J_{max} = \frac{\sigma \cdot \dot{\varepsilon}}{2} \tag{10}$$

For nonlinear dissipater, the efficiency of power dissipation $\eta$ can be defined as:

$$\eta = \frac{J}{J_{max}} = \frac{2m}{m+1} \tag{11}$$

$\eta$ is a dimensionless parameter, which indicates the relationship between the energy dissipated by microstructural evolution and the total energy dissipated during the hot deformation. Essentially, the parameter $\eta$ reflects the microstructural deformation mechanism of the workpiece in the corresponding range of deformation temperature and strain rate. The variation of $\eta$ with temperature and strain rate at a constant strain constitutes the power dissipation map [24]. In the process of plastic deformation, the formation of structural defects (such as voids, etc.,) and the occurrence of structural changes (such as dynamic recovery, dynamic recrystallization, etc.,) both dissipate power [25]. Therefore, combined with the metallographic microstructure, the power dissipation map can analyze the deformation mechanism at different deformation conditions. It should be noted that a significant value of $\eta$ in the power dissipation map does not necessarily mean better workability of the metal, because the value of $\eta$ may also be significant in some flow instability domains. Therefore, it is necessary to determine the processing instability domain of the experimental steel.

According to the principle of the maximum rate of entropy production, the flow instability will occur once the material meets the following conditions during hot deformation [26]:

$$\frac{dD(\dot{\varepsilon})}{d\dot{\varepsilon}} < \frac{D(\dot{\varepsilon})}{\dot{\varepsilon}} \tag{12}$$

where $D(\dot{\varepsilon})$ is the dissipation function at a given temperature. $D(\dot{\varepsilon})$ is equal to the power co-content, and Equation (12) can be rewritten as:

$$\frac{dJ}{d\dot{\varepsilon}} < \frac{J}{\dot{\varepsilon}} \tag{13}$$

The above equation can be further derived as:

$$\frac{\partial \ln J}{\partial \ln \dot{\varepsilon}} < 1 \tag{14}$$

By taking the logarithm on both sides of Equation (9), the new equation can be expressed as:

$$\ln J = \ln\left(\frac{m}{m+1}\right) + \ln \sigma + \ln \dot{\varepsilon} \tag{15}$$

Then, after taking the derivative of $\ln \dot{\varepsilon}$ on both sides of Equation (15), the following equation can be obtained:

$$\frac{\partial \ln J}{\partial \ln \dot{\varepsilon}} = \frac{\partial \ln\left(\frac{m}{m+1}\right)}{\partial \ln \dot{\varepsilon}} + \frac{\partial \ln \sigma}{\partial \ln \dot{\varepsilon}} + 1 \tag{16}$$

By integrating Equations (8), (13) and (16), the corresponding Prasad criterion can be derived as follows [27]:

$$\xi(\dot{\varepsilon}) = \frac{\partial \ln\left(\frac{m}{m+1}\right)}{\partial \ln \dot{\varepsilon}} + m = \frac{\partial \log\left(\frac{m}{m+1}\right)}{\partial \log \dot{\varepsilon}} + m < 0 \tag{17}$$

The instability parameter $\xi(\dot{\varepsilon})$ is a function of deformation temperature and strain rate [28]. By calculating the value of $\xi(\dot{\varepsilon})$ at different deformation conditions, and plotting the contour map of $\xi(\dot{\varepsilon})$ on a two-dimensional diagram composed of strain rate and temperature, the instability map can be constructed. A superimposition of the instability map over the power dissipation map gives a processing map [24].

## 4. Results and Discussion

### 4.1. Elimination of Element Segregation in As-Cast Microstructure

The metallographic microstructure of the as-cast S31254 SASS is shown in Figure 1a. A large number of precipitates are distributed in the interdendritic space. EDX analysis results confirm that the contents of Cr and Mo elements in the precipitates are significantly higher than those in the matrix. According to the results of the corresponding works, the precipitate is σ phase [29]. Figure 1b shows the SEM image of the as-cast microstructure. To evaluate the degree of element segregation, EDX surface scanning analysis is carried on the area within the green box in the figure. The results are shown in Figure 1c. It presents that the interdendritic positions have pronounced enrichment of Mo and S elements, accompanied by relative depletion of Fe and Ni elements. By comparing the nominal content of chemical elements in the steel in Table 1, it can be inferred that the directional diffusion of Mo element from dendrite to interdendritic space occurs during the solidification process. It provides a favorable condition for the precipitation of the Mo-rich σ phase, resulting in severe Mo element segregation in the as-cast microstructure.

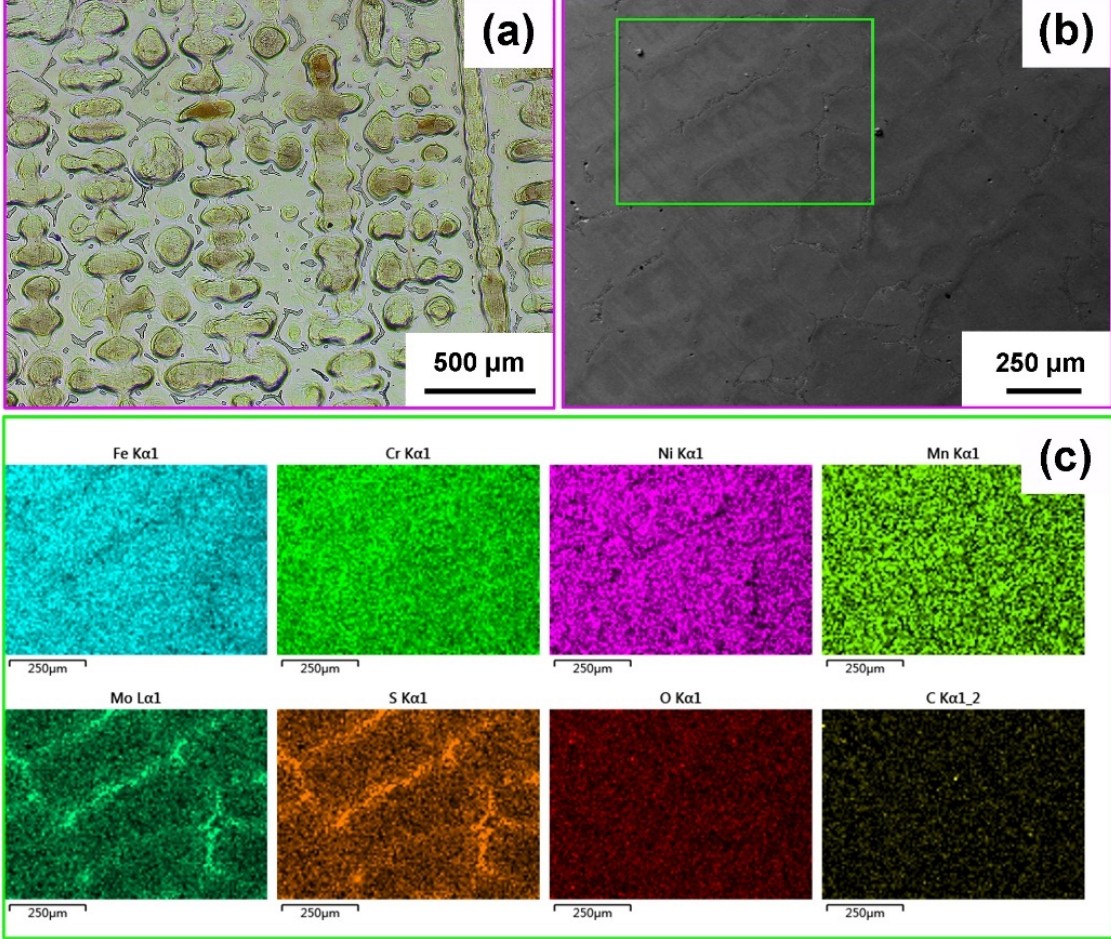

**Figure 1.** The element segregation analysis results, (**a**) metallographic microstructure of the as-cast S31254 SASS, (**b**) the position of the surface scan analysis in the SEM image of the as-cast microstructure, (**c**) the results of surface scanning analysis.

The segregation of Mo causes uneven microstructure and reduces the stability of deformative microstructure. So, an annealing treatment is required to eliminate element segregation before hot working. Figure 2 demonstrates the evolutional curve of the segregation of Mo after annealing for different durations. The coefficient *K* in the plot is defined as the ratio of the content of Mo in the interdendritic space to that in the dendrite. The value of *K* gradually decreases with the prolongation of annealing duration. When the annealing duration reaches 35 h, σ phases in the as-cast specimens are entirely dissolved. Until the annealing duration is 70 h, the value of *K* is 1.05, which is close to a uniform state of element distribution. By this time, the size of the austenitic grains have reached the mm level.

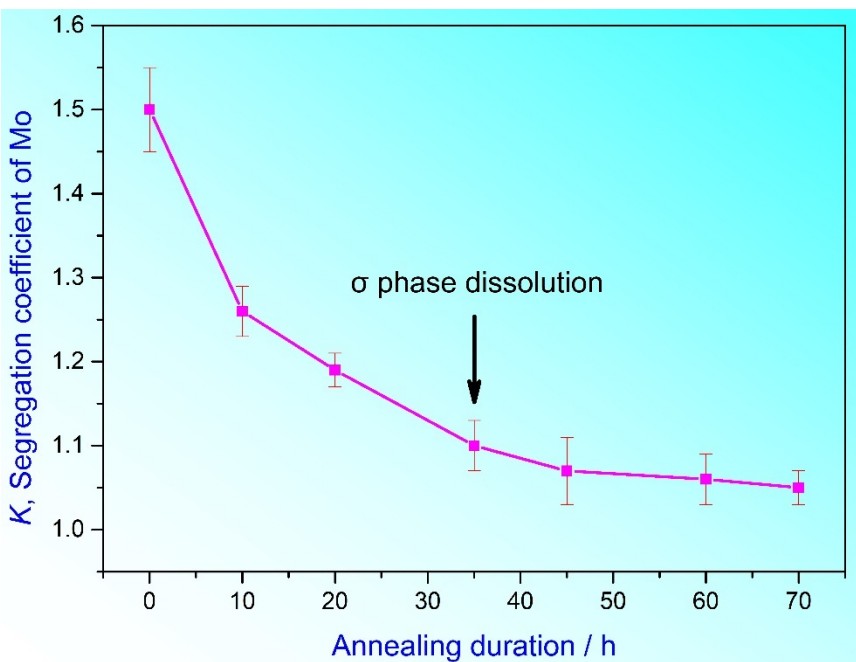

**Figure 2.** The relationship between the segregation coefficient of Mo element (*K*) and the annealing duration.

*4.2. Stress–Strain Relationship during Hot Compression*

When an alloy specimen is deformed at high strain rates, the dissipation of the instantaneous power will cause a temperature rise due to the effect of adiabatic heating [30,31]. According to the actual temperature measured by a thermocouple attached to the compressive specimen, it is found that the maximum temperature rise of specimen deformed at the strain rate of $1 \, \text{s}^{-1}$ is 16 °C and the maximum temperature rise at $10 \, \text{s}^{-1}$ is 24 °C. While the temperature variation results from deformation is negligible at low strain rate. Therefore, the stress–strain data in this test need to take into account the effect of temperature variation, especially at the strain rates of 1 and $10 \, \text{s}^{-1}$. The temperature-corrected flow stress ($\sigma_T$) can be expressed as follows [32]:

$$\sigma_T = \sigma_i e^{\frac{Q'}{R}\left(\frac{1}{T_D} - \frac{1}{T_R}\right)} \tag{18}$$

where $\sigma_i$ is the raw stress detected in the test (MPa), $Q'$ is a constant coefficient ($\text{J} \cdot \text{mol}^{-1}$), $R$ is the universal gas constant ($8.314 \, \text{J} \cdot \text{mol}^{-1} \cdot \text{K}^{-1}$), $T_D$ is the specified temperature of specimen (K), and $T_R$ is the actual temperature of specimen (K).

The coefficient $Q'$ can be obtained by fitting the slope of the dependence of the yield stress on the temperature at a strain of 0.3. Figure 3a,b exhibits the results of the fitting and the values of $Q'$ are $53 \pm 4 \, \text{kJ} \cdot \text{mol}^{-1}$ for the strain rate of $1 \, \text{s}^{-1}$ and $46 \pm 4 \, \text{kJ} \cdot \text{mol}^{-1}$ for the strain rate of $10 \, \text{s}^{-1}$ in this experiment. The temperature-corrected stress varying with strain at the strain rate of 1 and $10 \, \text{s}^{-1}$ are presented in Figure 3c,d, respectively. The

increment of flow stress resulted from adiabatic heating decreases with the increase of the deformation temperature. Moreover, the increase of the strain rate leads to a more pronounced effect of adiabatic heating on the flow stress.

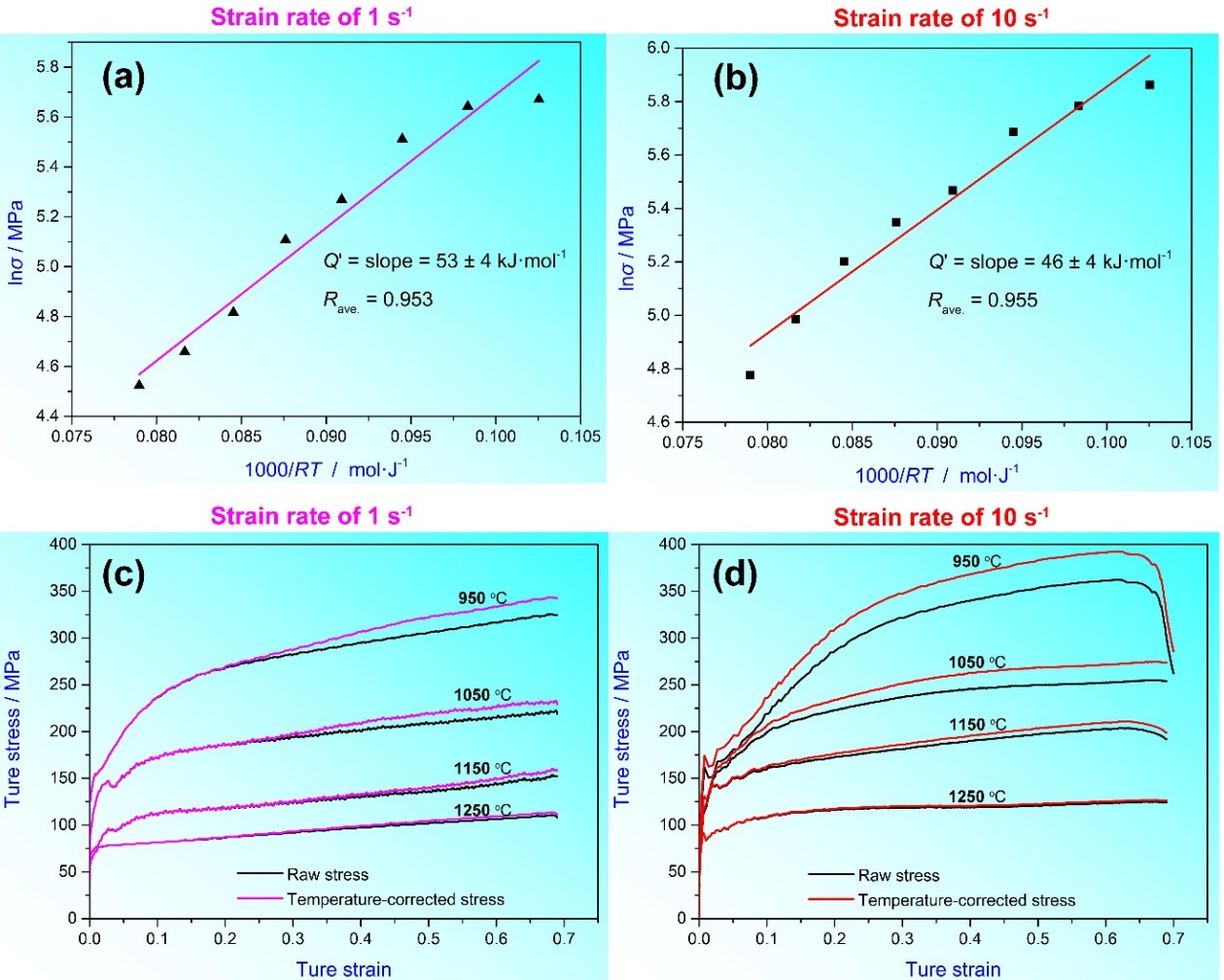

**Figure 3.** The dependence of $\ln\sigma$ on $1/T$ at a strain rate of 1 (**a**) and 10 s$^{-1}$ (**b**). The temperature-corrected stress varying with strain at the strain rates of 1 (**c**) and 10 s$^{-1}$ (**d**).

The true stress–strain curves at various deformation conditions are illustrated in Figure 4. Each curve is measured at least three times, and a representative selection is presented here. The flow stress enhances with the increase of the strain rate in the same deformation temperature; meanwhile, in the same strain rate, the flow stress increases with the decrease of the deformation temperature. It proves that S31254 SASS is a type of material with positive strain rate sensitivity. The energy required for plastic deformation improves with the increase of strain rate or the decrease of deformation temperature. In the reported studies on fine-grained S31254 SASS, the flow stress could reach the peak stress at a particular strain, then decreased gradually [33]. This resulted from the interaction of the hardening and softening effects of the metal during hot deformation. With the progress of dynamic recovery and dynamic recrystallization, the hardening effect gradually weakened, and the flow stress gradually decreased after reaching a peak value. Then the flow curve reached a steady stage. However, the flow curves in this study show some differences. The flow stress of several curves do not climb to peak within the true strain range of 0–0.7, but increase monotonically. It indicates that the coarse-grained S31254 SASS has particularity in the evolution of the hot deformation microstructure.

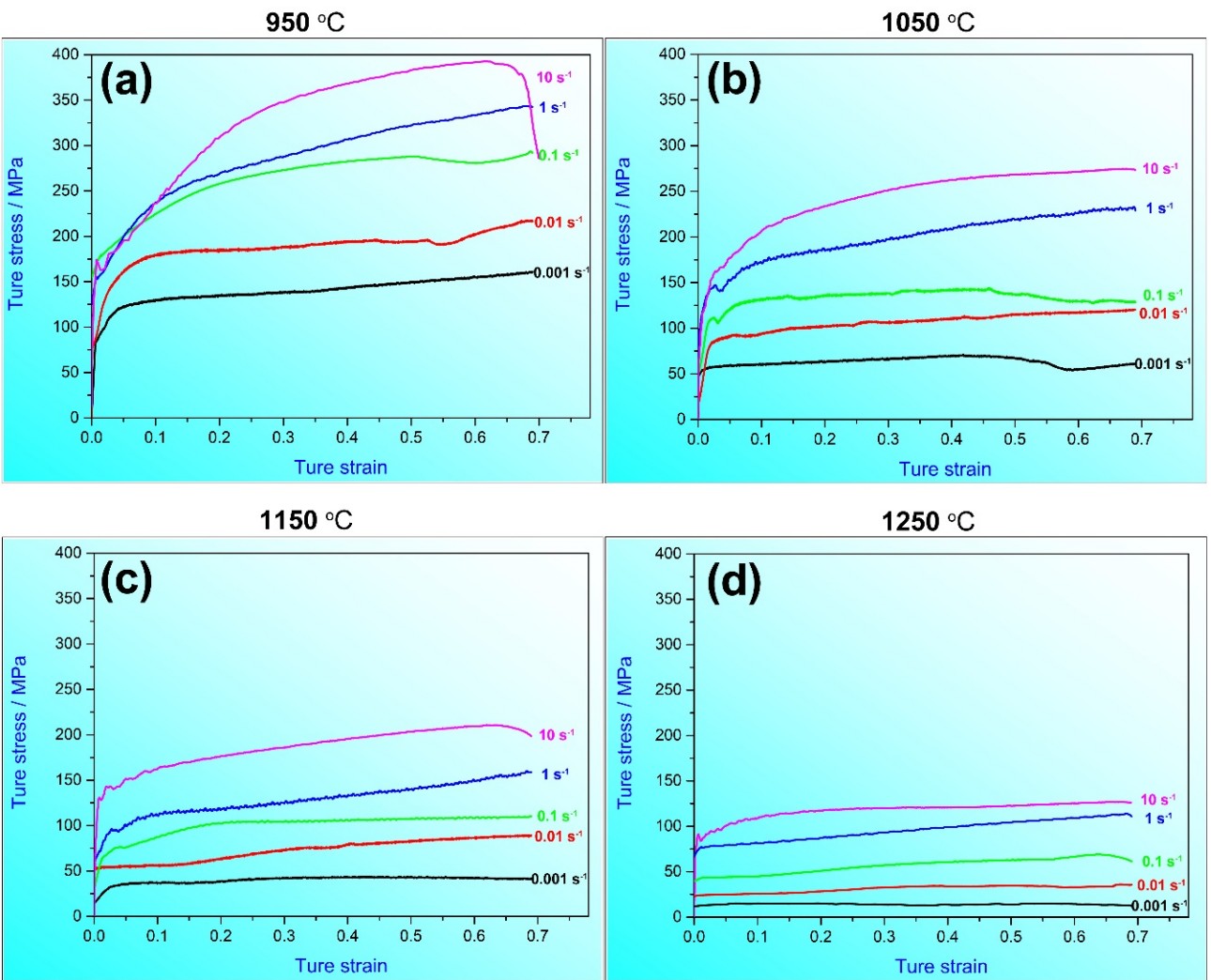

**Figure 4.** The true stress–strain curves of S31254 SASS specimens compressed at various temperatures of 950 °C (**a**), 1050 °C (**b**), 1150 °C (**c**), and 1250 °C (**d**).

*4.3. Hot Deformation Constitutive Model*

The relationship plots between $\log \dot{\varepsilon}$ and $\log[\sinh(\alpha\sigma)]$ in different temperatures are illustrated in Figure 5a. By linearly fitting the data points and calculating the average value of the slopes of all the fitted straight lines, the value of the stress exponent *n* can be calculated to be 3.8 ± 0.37. The value of *n* lies close to 3, indicating that the mode of dislocation movement is dominated by glide and climb in the process of plastic deformation [7].

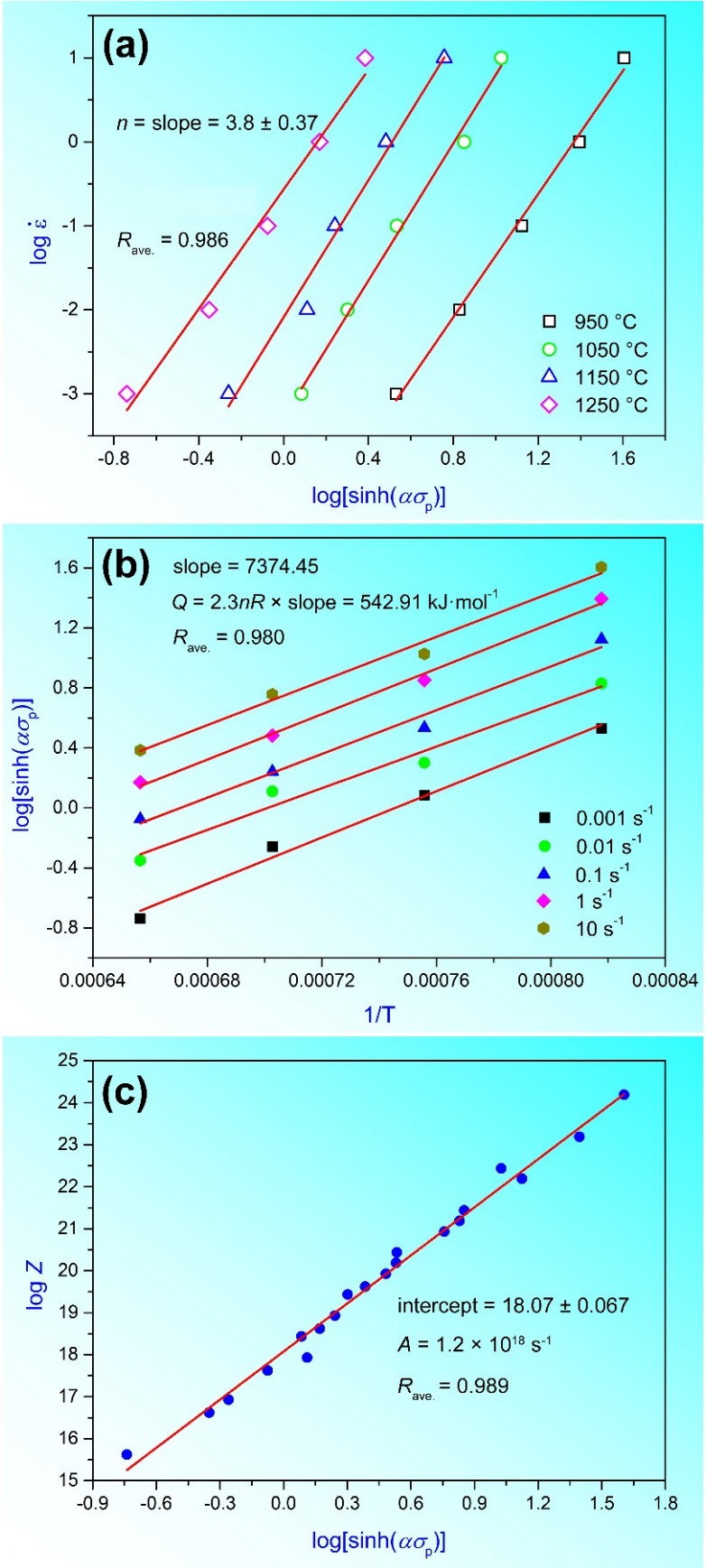

**Figure 5.** The plot between $\log\dot{\varepsilon}$ and $\log[\sinh(\alpha\sigma)]$, where $\dot{\varepsilon}$ is in s$^{-1}$ (**a**), the plot between $\log[\sinh(\alpha\sigma)]$ and $1/T$, where $T$ is in K (**b**), the plot between $\log Z$ and $\log[\sinh(\alpha\sigma)]$ (**c**).

By inserting the values of $\alpha$ and $n$ into Equation (5), the relationship plots between $\log[\sinh(\alpha\sigma)]$ and $1/T$ at different strain rates can be drawn, as presented in Figure 5b. The value of activation energy $Q$ can be calculated by averaging the slope value of the fitted straight line at each strain rate. The $Q$ of the coarse-grained S31254 SASS is calculated to be 542.91 kJ·mol$^{-1}$. This value is close to some research results about fine-grained S31254 SASS, and is significantly higher than a few austenitic stainless steels [34–36]. It indicates that the coarse-grained S31254 SASS has greater hot deformation resistance.

Take the logarithm on both sides of the Equation (3) which is substituted by Equation (2). The relationship between the $Z$ parameter and the flow stress can be described as Equation (19).

$$\log Z = \log A + n \log[\sinh(\alpha\sigma)] \tag{19}$$

By taking the values of $\alpha$, $n$, and $Q$ into Equation (19), the value of $\log Z$ corresponding to $\log[\sinh(\alpha\sigma)]$ at different deformation temperatures and different strain rates can be obtained, and the results of the linear fitting are presented in Figure 5c. According to the intercept of the fitted straight line, the value of the material constant $A$ is calculated to be $1.2 \times 10^{18}$ s$^{-1}$.

Consequently, taking the values of parameters $\alpha$, $n$, $Q$, and $A$ into Equations (1) and (2), the hot deformation constitutive equation of coarse-grained S31254 SASS can be established, as expressed in the following:

$$\begin{cases} \sigma = \dfrac{1}{0.012}\left[\sinh^{-1}\left(\dfrac{Z}{1.2\times 10^{18}}\right)^{\frac{1}{3.8}}\right] \\ Z = \dot{\varepsilon}\, exp\left(\dfrac{542,910}{RT}\right) \end{cases} \tag{20}$$

This constitutive equation has reference significance for practitioners in the steel industry to deal with the problems related to the hot working of S31254 SASS.

*4.4. Processing Map and the Optimum Working Conditions*

The processing maps of coarse-grained S31254 SASS at strains of 0.2, 0.4, and 0.6 are shown in Figure 6a–c, respectively. The contour lines represent the efficiency of power dissipation in percentage, and shaded areas represent instability domains. With the increase of strain, the instability domain at high temperatures will gradually enlarge. The processing maps exhibit two different regions with relatively high efficiencies of power dissipation. The first region occurs in the deformation temperature range of 1020–1080 °C and the strain rate range of 0.001–0.005 s$^{-1}$. It is only evident in the map of Figure 6c. The second region occurs in the deformation temperature range of 1200–1250 °C and the strain rate range of 0.001–0.5 s$^{-1}$. This region represents the peak efficiency of power dissipation at each map, and with the increase of the strain, the efficiency of power dissipation gradually increases. When the strain is 0.2 or 0.4, no flow instability domain exists in this region; nevertheless, when the strain reaches 0.6, the flow instability domain appears in the low strain rate range (0.001–0.01 s$^{-1}$). Therefore, according to the analysis of processing maps, the two optimum regions for the hot working of coarse-grained S31254 SASS are deformation temperature range of 1020–1080 °C, strain rate range of 0.001–0.005 s$^{-1}$, and deformation temperature range of 1200–1250 °C, strain rate range of 0.01–0.5 s$^{-1}$.

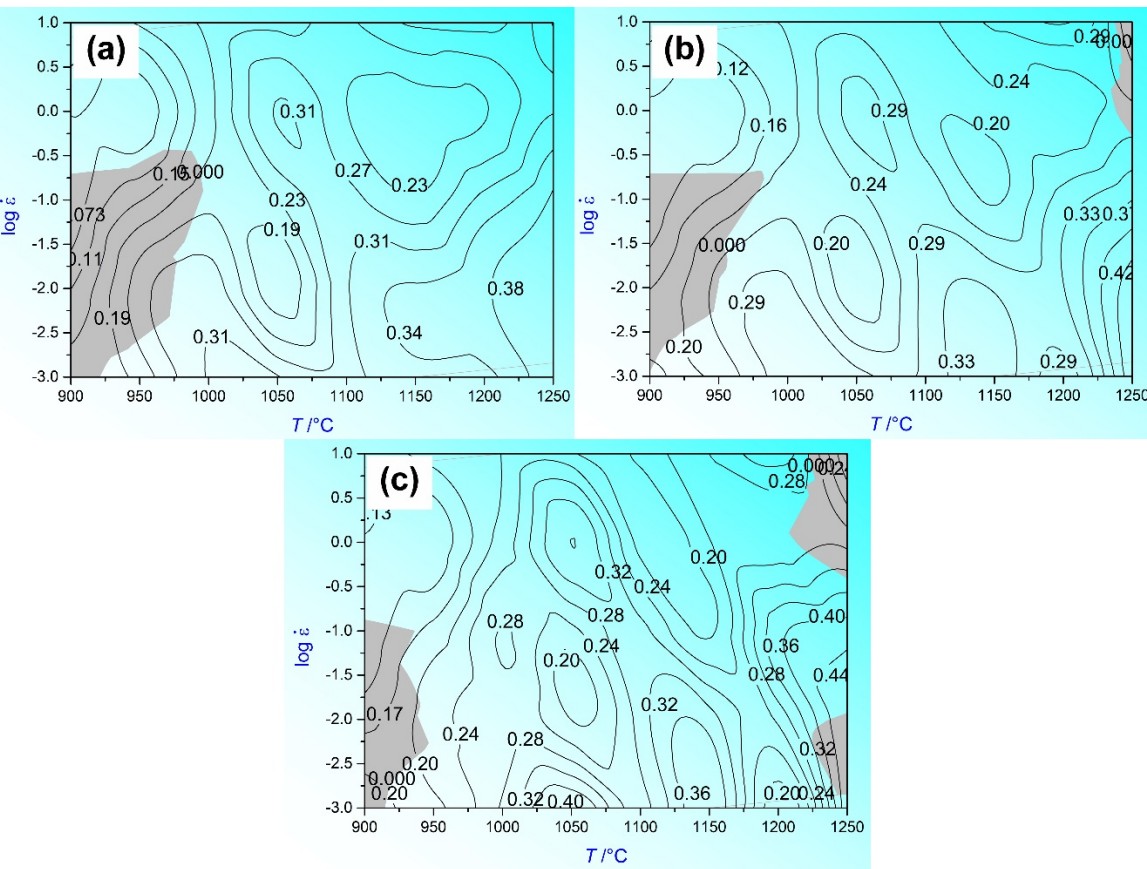

**Figure 6.** The processing maps of coarse-grained S31254 SASS at strain of 0.2 (**a**), 0.4 (**b**), and 0.6 (**c**).

### 4.5. Characteristics of Hot Deformation Microstructures

The EBSD analysis of the compressed specimens at different conditions is illustrated in Figure 7. The recrystallization maps of the specimens compressed at 0.1 s$^{-1}$ with different temperatures of 950 °C, 1050 °C, and 1150 °C are presented in Figure 7a–c. Since the grain size grows to the level of millimeter, the fraction of grain boundaries which are the preferred sites for the nucleation of recrystallized grains is low, resulting in the difficulty in dynamic recrystallization [37]. The microstructures in Figure 7a–c are dominated by deformed austenitic grains and substructures, and the proportion of recrystallized grains is very low. Moreover, many band-like structures which are formed by the bending of grains during plastic deformation can be observed in the deformed grains. The recrystallization maps of the specimens compressed at 1 s$^{-1}$ with different temperatures of 1050 °C, 1150 °C, and 1250 °C are presented in Figure 7d–f. The microstructures of the specimens whose deformation temperatures are lower than 1150 °C are still composed of deformed austenitic grains and substructures, and the dynamic recrystallization phenomenon is still insignificant. Simultaneously, the fraction of band-like structure in the deformed grains is higher than that in the compressed specimens at the strain rate of 0.1 s$^{-1}$. While the deformation temperature reaches 1250 °C, the recrystallization phenomenon occurs obviously. The recrystallized grains are distributed along with the band-like structures, and the grain size is significantly refined. It should be noted that no σ phase is precipitated in all compressed specimens, which is different from the studies about fine-grained S31254 SASS.

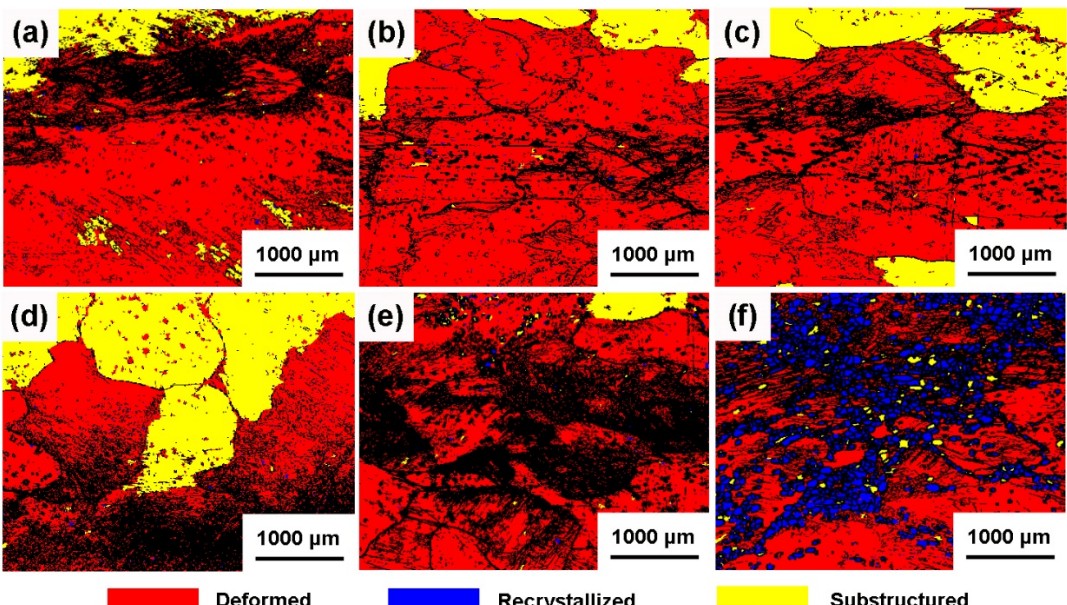

**Figure 7.** The EBSD recrystallization maps (BC + DefRex + GB) of specimens compressed at $0.1 \text{ s}^{-1}$ with different temperatures of 950 °C (**a**), 1050 °C (**b**), 1150 °C (**c**), and compressed at $1 \text{ s}^{-1}$ with different temperatures of 1050 °C (**d**), 1150 °C (**e**), 1250 °C (**f**). The red area in the figure represents deformed grain, the yellow area represents substructure, the blue area represents recrystallized grain, and the black area represents grain boundary or band-like structure where the stress is concentrated.

A higher strain rate means the specimen will experience a more significant applied force. So, the energy input to the specimen is higher, which results in a higher fraction of band-like structure in the specimens compressed at the strain rate of $1 \text{ s}^{-1}$. In addition, through a comprehensive analysis of the processing maps in Figure 6 and the EBSD recrystallization maps in Figure 7, it could be concluded that the first optimum region for hot working lies in the non-recrystallized domain. The plastic deformation mechanism of the microstructure is dominated by dynamic recovery. The microstructure shows dynamic recrystallization phenomenon evidently when the specimens meet the conditions of the second optimum region in the processing map. Dynamic recrystallization has a significant improvement on the stability of the deformed structure. Thus, the efficiency of power dissipation in the second optimum region in the processing map is higher. A band-like structure has a higher energy level within the deformed austenitic grain. Thus, it is the preferred position for nucleation of recrystallized grain here. However, the occurrence of recrystallization additionally requires a high deformation temperature to provide the nucleation driving force. By summarizing the above experimental results, it can be concluded that only when the temperature reaches 1250 °C that the recrystallized grains have sufficient driving force to nucleate and grow. Consequently, the fraction of dynamic recrystallized grains increases significantly.

The annealing for 70 h seems too long, causing the grains to overgrow. However, the specimens do not precipitate any σ phase in the hot compression experiment. Considering the negative influence of σ phase on plastic deformation, a long-term annealing is beneficial for the workability of S32154 SASS. In some reports, it was believed that the σ phase tended to precipitate rapidly in 1050 °C, which should be avoided during hot working [38]. However, no precipitation of the σ phase was detected near this temperature in the experiment, indicating that long-term annealing significantly reduced the precipitating tendency of the σ phase. Consequently, it is of great significance to clarify the optimum range of the hot working parameters for the actual manufacturing of the coarse-grained S31254 SASS.

### 4.6. Evaluation of Microstructural Stability during Hot Tensile Tests

During hot working (such as forging or hot rolling), different locations of the ingot experience different types of stress. The part of the ingot parallel to the pressure-bearing surface is mainly affected by compression stress. In contrast, the center of the side perpendicular to the pressure-bearing surface is affected by tension stress. The hot tensile tests were carried out to determine the tensile strength and reduction of the area. So, the microstructural stability of the specimen under the action of tension stress can be evaluated [39]. The hot tensile curves of the specimens at the strain rate of 0.5 and 5 s$^{-1}$ are presented in Figure 8a,b, respectively. Elongation decreases with increasing deformation temperature at both strain rates. The state of failed tensile specimens is shown in Figure 9a. It can be seen that, except for the specimen tested at the temperature of 1250 °C and strain rate of 5 s$^{-1}$, the rest have distinct necking phenomena. According to the plot of the tensile properties in Figure 9b, the tensile strength decreases monotonically with the increase of the deformation temperature. This rule is the same as the evolution of peak stress in the hot compression test. The variation of the reduction of area with the deformation temperature is comparable at different strain rates. It gradually improves with the increase of the temperature from 950 °C to 1150 °C, but turns to decrease at 1250 °C. The correlation between the reduction of area and the strain rate is not significant in temperature range of 950–1150 °C. However, at 1250 °C, the reduction of area of the specimen tested at the rate of 5 s$^{-1}$ is significantly lower than that at the rate of 0.5 s$^{-1}$.

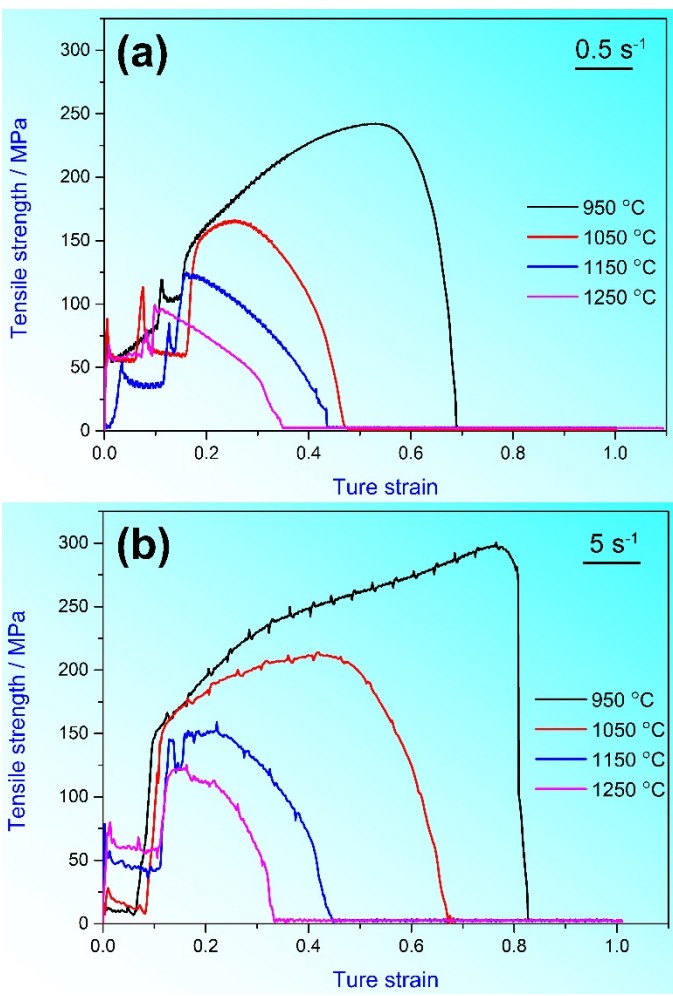

**Figure 8.** The hot tensile curves of the specimens at the strain rate of 0.5 (**a**) and 5 s$^{-1}$ (**b**).

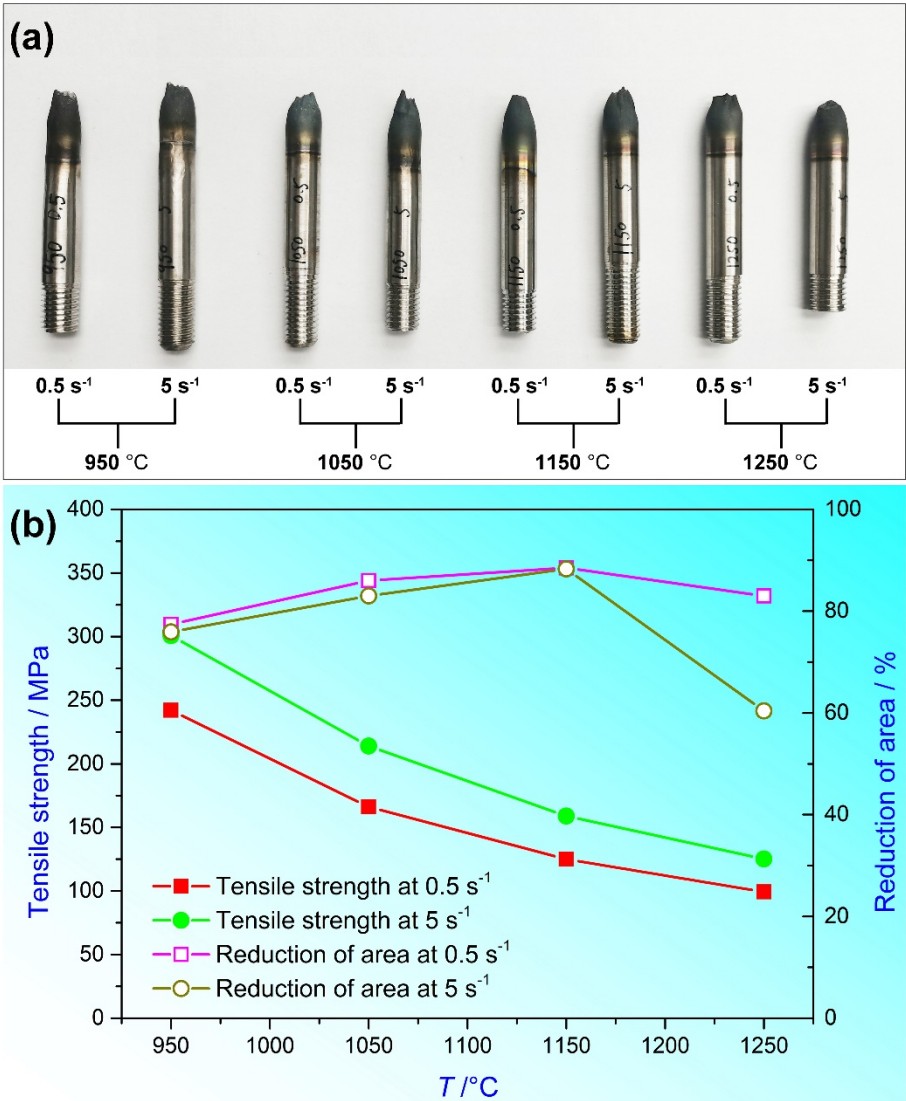

**Figure 9.** The state of the failed hot tensile specimens (**a**) and the evolution of tensile properties with different tensile parameters (**b**).

The SEM fractography of the failed tensile specimen are demonstrated in Figure 10. For the specimens tested in the temperature range of 950–1150 °C, numerous dimples can be found on the fracture surfaces no matter the strain rate. It presents a typical feature of ductile fracture mode. The size and number of dimples differ little between specimens. The fracture surface of the specimen tested at 950 °C is smooth; nevertheless, those of the specimens tested in the temperature range of 1050–1250 °C are relatively rough. For the specimen tested at 1250 °C, when the strain rate is 0.5 s$^{-1}$, the fracture mode is still ductile. However, when the strain rate increases to 5 s$^{-1}$, many tearing edges appear, and the number of dimples obviously reduces. It indicates that the plastic deformation ability of the microstructure worsens.

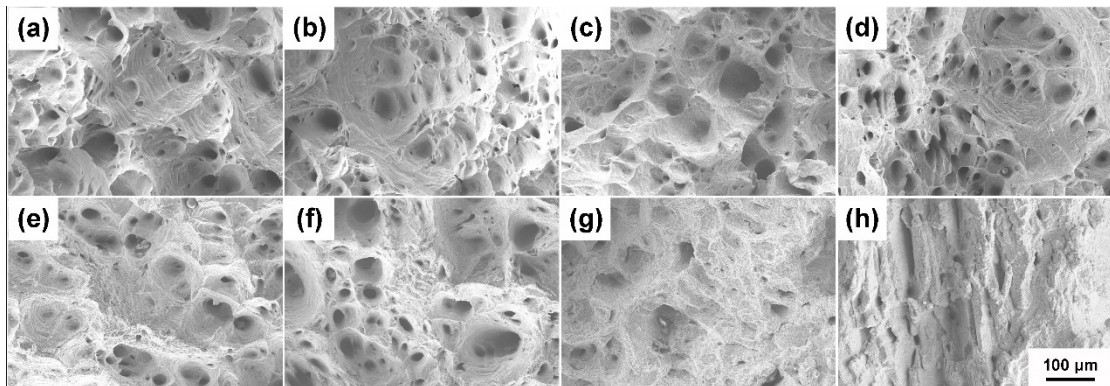

**Figure 10.** The SEM fractographs of the specimens tested in 950 °C (**a**), 1050 °C (**b**), 1150 °C (**c**), 1250 °C (**d**) at strain rate of 0.5 s$^{-1}$, and tested in 950 °C (**e**), 1050 °C (**f**), 1150 °C (**g**), 1250 °C (**h**) at strain rate of 5 s$^{-1}$.

To find out the reason for the sudden decrease in hot ductility of the specimen at a temperature of 1250 °C and strain rate of 5 s$^{-1}$, the equilibrium phase composition diagram of the experimental steel is plotted by thermodynamic calculation in the range of 800 to 1500 °C, as shown in Figure 11a. The austenite and σ phase are the main components in the temperature range of 950–1150 °C, and the mass fraction of σ phase decreases with the increase in temperature. It explains the variation of hot ductility in this temperature range. When the temperature is 1250 °C, although the σ phase is completely dissolved, the mass fraction of δ-ferrite phase is about 10%, which means the transformation from austenite to δ-ferrite phase will occur. The δ-ferrite phase is preferentially formed at the grain boundary of austenite, resulting in intergranular fracture failure during hot tensile. In addition, the formation temperature of the liquid phase was determined to be 1365 °C by the DSC test in Figure 11b. Due to the effect of adiabatic heating, the actual temperature of the specimen tensiled at 1250 °C and 5 s$^{-1}$ was 1267 °C. This temperature is close to the liquidus temperature of the experimental steel, which may lead to the local melting of the austenitic grain boundary [40]. It has a fatal influence on the hot plasticity of the experimental steel.

Based on the hot tensile properties and the fracture feature, the stability of the microstructure under the action of tension stress is high when the deformation temperature is between 950 °C and 1150 °C. Thus, the tendency of cracking in the center of the side is low when the ingot is hot worked in this temperature range. Once the temperature increases to 1250 °C, the strain rate has to be controlled. At this time, if the strain rate is higher than 0.5 s$^{-1}$, the plastic deformation ability of the specimen will decrease, and the stability of microstructure will deteriorate. In the above chapter, it has been proposed that the microstructure of the specimen has good plastic deformation stability in the temperature range of 1200–1250 °C and the strain rate range of 0.01–0.5 s$^{-1}$. The analysis of the hot tensile fracture feature can further prove that, when the specimen is hot worked in these deformation parameter ranges, the tendency of surface cracking is low, and the hot working performance is excellent.

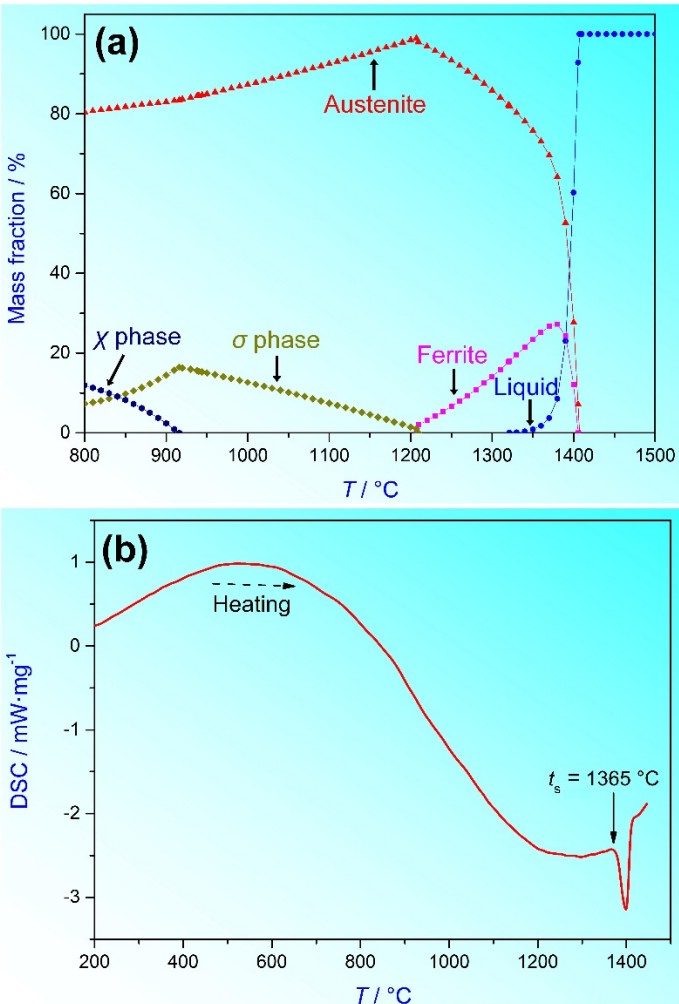

**Figure 11.** The equilibrium phase composition of the experimental steel in the temperature range of 800–1500 °C (**a**) and the curve of DSC test (**b**).

## 5. Conclusions

In this study, the elemental segregation of the as-cast S31254 SASS has been eliminated by a long-term annealing. The hot compression and hot tensile tests have been conducted to evaluate the microstructural stability during hot working. The following conclusions can be drawn:

1. Resulting from the solidification segregation and the precipitation of the Mo-rich σ phase, the Mo element in the as-cast microstructure was significantly enriched in the interdendritic space. An annealing treatment at 1220 °C for 70 h could eliminate the segregation of the Mo element, and ensured that the σ phase did not precipitate in the process of hot deformation.

2. The hot compression stress–strain curves showed that S31254 SASS was a type of material with positive strain rate sensitivity. The energy required for plastic deformation improved with the increase of strain rate and the decrease of deformation temperature. The hot deformation activation energy was calculated to be 542.91 kJ·mol$^{-1}$ through the regression analysis of hyperbolic-sine function. The constitutive equation was established, and its expression was:

$$\begin{cases} \sigma = \dfrac{1}{0.012}\left[\sinh^{-1}\left(\dfrac{Z}{1.2\times10^{18}}\right)^{\frac{1}{3.8}}\right] \\ Z = \dot{\varepsilon}\, exp\left(\dfrac{542{,}910}{RT}\right) \end{cases}$$

3. The processing maps consisting of the power dissipation map and the instability map were constructed, and two optimal hot working parameters ranges were clarified. That was, the deformation temperature range was 1020–1080 °C, the strain rate range was 0.001–0.005 s$^{-1}$, and the deformation temperature was 1200–1250 °C, the strain rate range was 0.01–0.5 s$^{-1}$.

4. Due to the low fraction of grain boundaries, it was difficult for recrystallized grains to nucleate. So, the main deformation mechanism of coarse-grained S31254 SASS was dynamic recovery. Numerous band-like structures existed in the deformed austenitic grains in the process of hot deformation. However, when the deformation temperature reached 1250 °C, recrystallized grains began to nucleate and grow along with the band-like structure.

5. When the deformation temperature was between 950 °C and 1150 °C, the microstructural stability of coarse-grained S31254 SASS specimen under tension stress was excellent. However, when the deformation temperature increased to 1250 °C and the strain rate was 5 s$^{-1}$, the microstructural stability deteriorated. The formation of δ-ferrite phase and local melting of austenitic grain boundaries are the reasons for this phenomenon.

**Author Contributions:** Conceptualization, J.X. and C.L.; methodology, J.X.; software, J.X.; validation, C.L. and S.W.; writing-original draft preparation, J.X.; writing-review and editing, J.X.; formal analysis, J.X. and X.Z.; investigation, J.X.; resource, S.W. and Y.S.; supervision, A.L.; project administration, J.X. and C.L.; funding acquisition, J.X. All authors have read and agreed to the published version of the manuscript.

**Funding:** This study was funded by Fundamental Research Program of Shanxi Province (No. 20210302124044).

**Institutional Review Board Statement:** Not applicable.

**Informed Consent Statement:** Not applicable.

**Data Availability Statement:** Data sharing is not applicable.

**Acknowledgments:** We thank Fundamental Research Program of Shanxi Province (No. 20210302124044) for the support of this study.

**Conflicts of Interest:** The authors declare no conflict of interest.

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
