# Peer review of "Microstructural Evolution and Stability of Coarse-Grained S31254 Super Austenitic Stainless Steel during Hot Deformation"

_metals, doi:10.3390/met12081319_

Round 1

Reviewer 1 Report

In the paper "Microstructural Evolution and Stability of Coarse-Grained S31254 Super Austenitic Stainless Steel During Hot Deformation", the authors have investigated the microstructure evolution and constructed a constitutive model of hot deformation behaviour of the S31254 steel. The authors have also determined the flow stability regions using processing maps. The high value of the hot ductility of the steel was approved for the temperature range of 950 – 1150 °C using tension tests. The paper presents interesting results. However, some parts of the manuscript are needed to be modified accordingly following comments:

1.                 Part of the stress-strain curves has a strange view. What was the reproducibility of the compression test? It is recommended to repeat a few times test at 950 °C and 10 s-1.

2.                 Аdiabatic heating during the deformation at high strain rates may significantly influence the true stress – true strain curves [10.1016/j.jallcom.2018.08.010, 10.1134/S0031918X14080031]. Did the authors consider this fact?

3.                 What was the strain value for the constitutive model construction? The value of the effective activation energy has a meaning if it was determined for the peak stress value or the steady-state value. Did the authors consider this fact?

4.                 The authors did not describe the reason for the decreasing the hot ductility of the steel at 1250 °C and 5 s-1. It seems that the local melting at grain boundaries proceeds due to adiabatic heating above solidus temperature during the tension test. The authors should measure the solidus temperature using the DTA technique and provide the measured temperature during the tension test.

5.                 Minor corrections:

-                     It is recommended to add the average relative error to Figure 4c in addition to the Pearson’s coefficient.

-                     The number of digits in the model’s coefficients should be decreased accordingly to the error of their determination.

Reviewer 2 Report

The article is devoted to the study of the behavior of super austenitic stainless steel during heating and hot deformation. The effect of Mo on structure formation during hot deformation of steel grade S31254 was studied. The study used modern research methods and equipment. However, when reviewing, I had a few questions and comments that, I hope, will help the authors improve the quality of the paper submitted for publication.
1. How is the graph in Figure 2 obtained? How was the segregation coefficient calculated?
2. Fractograms (Fig. 8) require a more detailed explanation. The photo of 1250 °C (h) is especially worth paying attention to. It should be explained in more detail what is a sharp loss of plastic and strength properties at this temperature (Fig. 7).
3. In fig. 6 there is a confusion with the designations (d) and (g). You need to correct mistakes.
4. Why didn't you bring the true stress-strain curves in tension? I think it's worth adding these results to the paper. These results will be very useful to readers.
5. In addition to recrystallization maps (Fig. 6), it would be useful to show a photo of the microstructure without filters, i.e. in its original state (as done in the article https://doi.org/10.3390/ma15124057). I recommend that you familiarize yourself with the indicated article, since similar studies are carried out in it, and perhaps it will help to describe better and explain the results obtained.

Reviewer 3 Report

In this article, the authors analyzed the microstructural evolution and stability of corase-grained austenitic stainless steel during hot forming. They analyzed the effect of alloying elements on the evolution of phase dissolution, the stress-strain behavior, and resulting process maps, as well as the mechanical behavior of the materials at different temperatures and strain rates.

The work in general is interesting and can be useful for the research community. I have a few of the following suggestions to improve the presentation and clarity of the work for future readers:

- The language of the article needs significant improvement; some sentences do now make sense i.e., page 1, line 52 (uniformize), page 2, line 86 (reached mm level), and others, such incidents

- The problem statement definition at the end of the introduction section needs significant improvement. page 2, line 72-75

- The research objectives at the end of the introduction section need significant improvement. Page 2, lines 76-80

- I believe that there would be a separate numerical model section before the results and discussion section where the authors can include the models and equations presented later in 3.3 and 3.4 as they do not belong in the results section.

- In figure 3, all figures should be on the same scale (y-axis), this will help in easier comparison of plots for different temperatures.

- I do not understand Figure 6, the authors say that it is an EBSD map, but I do not see how it has been structured. I could not see any IPF colors here. Do different colors signify different zones? how were they identified?

- Please cite the most recent work in this area on the effect of microstructure on the mechanical properties in the introduction of the article:

* https://doi.org/10.3139/146.111900

* https://doi.org/10.3390/cryst10030221

Round 2

Reviewer 1 Report

The authors have answered previous comments and significantly improved the manuscript. The paper may be accepted for publication. However, added to the manuscript text should be checked for grammar errors.

Reviewer 2 Report

The authors made corrections and additions to the article. The authors of the paper responded to my comments. I recommend that this article be published in this version.